# Less need for differentiation? Intestinal length of reptiles as compared to mammals

**Monika I. Hoppe**[1], **Carlo Meloro**[2], **Mark S. Edwards**[3], **Daryl Codron**[4], **Marcus Clauss**[1]*, **María J. Duque-Correa**[1]

**1** Clinic for Zoo Animals, Exotic Pets and Wildlife, Vetsuisse Faculty, University of Zurich, Zurich, Switzerland, **2** Research Centre in Evolutionary Anthropology and Palaeoecology, Liverpool John Moores University, Liverpool, United Kingdom, **3** California Polytechnic State University, San Luis Obispo, California, United States of America, **4** Department of Zoology and Entomology, University of the Free State, Bloemfontein, South Africa

* mclauss@vetclinics.uzh.ch

**Data Availability Statement:** All relevant data are within the paper and its Supporting Information files.

## Abstract

Although relationships between intestinal morphology between trophic groups in reptiles are widely assumed and represent a cornerstone of ecomorphological narratives, few comparative approaches actually tested this hypothesis on a larger scale. We collected data on lengths of intestinal sections of 205 reptile species for which either body mass (BM), snout-vent-length (SVL) or carapax length (CL) was recorded, transforming SVL or CL into BM if the latter was not given, and analyzed scaling patterns with BM and SVL, accounting for phylogeny, comparing three trophic guilds (faunivores, omnivores, herbivores), and comparing with a mammal dataset. Length-BM relationships in reptiles were stronger for the small than the large intestine, suggesting that for the latter, additional factors might be relevant. Adding trophic level did not consistently improve model fit; only when controlling for phylogeny, models indicated a longer large intestine in herbivores, due to a corresponding pattern in lizards. Trophic level effects were highly susceptible to sample sizes, and not considered strong. Models that linked BM to intestine length had better support than models using SVL, due to the deviating body shape of snakes. At comparable BM, reptiles had shorter intestines than mammals. While the latter finding corresponds to findings of lower tissue masses for the digestive tract and other organs in reptiles as well as our understanding of differences in energetic requirements between the classes, they raise the hitherto unanswered question what it is that reptiles of similar BM have more than mammals. A lesser effect of trophic level on intestine lengths in reptiles compared to mammals may stem from lesser selective pressures on differentiation between trophic guilds, related to the generally lower food intake and different movement patterns of reptiles, which may not similarly escalate evolutionary arms races tuned to optimal agility as between mammalian predators and prey.

## Introduction

There is a long tradition in ecomorphology of linking variation in digestive tract anatomy in vertebrates to the natural diets of the species, with the typical claim that herbivores have more

**Funding:** This study is part of Swiss National Science Foundation (http://www.snf.ch) project CRSII5_189970 / 1 (to MC). The funders had no role in study design, data collection and analysis, decision to publish, or preparation of the manuscript.

**Competing interests:** The authors have declared that no competing interests exist.

complex, more voluminous, and also longer gut sections than carnivores [1, 2]. Given the widespread acceptance of this assumption, the paucity of actual statistical tests is surprising. In mammals, for example, statistical associations between intestinal length and diet had not been demonstrated until recently [3].

In groundbreaking work on 40 reptile species, Lönnberg [4] compared the snout-vent length to intestine length, showing that herbivores had an intestinal length of 293% relative to the snout-vent length, whereas omnivore and carnivore species had 184% and 131%, respectively. These findings shaped our understanding of reptile intestinal macroanatomy [5–7]; they have, however, not been assessed statistically.

Based on visual comparisons and analyses of a few species, reptiles have shorter digestive tracts than mammals [8–10]; they have significantly less digestive tissue mass [11]. This is generally explained by their lower metabolism and lower food throughput compared to endotherms [2, 8]. A lesser food throughput might, however, reduce the pressure on the reptilian digestive tract to be strictly associated with the peculiarities of a specific diet. Together with a lesser dependence on movement and agility compared to endotherms [12, 13], there might be less pressure to streamline aspects of the body *bauplan*. This rationale has been used to explain the seeming absence of a difference in torso volume in a small dataset between faunivorous and herbivorous reptiles, when such a difference is evident in mammals [14]. Visual impressions of digestive tracts of faunivorous vs. herbivorous species support the notion that differences between reptile trophic groups may be less distinct than they are in mammals (Fig 1). Therefore, we collated intestinal length data for reptiles of known body mass, to assess general scaling relationship, the comparison with mammals, and the influence of trophic level on these measures.

## Materials and methods

Relevant publications were collated by first using the sources of Franz et al. [11] as a starting point, but also by performing a literature research in the three search engines Google scholar, PubMed, and Web of Science. Search terms included 'intestine'/'gastrointestinal tract', 'length'/'morphology', 'reptile'/scientific names of reptile genera. Reference lists, and 'cited by' lists, of publications thus identified were also used. Sources were only included if they provided the species, body mass (BM) or snout-vent-length (SVL) or carapace length (CL) of the individuals studied, as well as measurements of all or some of the gut sections (total intestine–TI, small intestine–SI, large intestine–LI including the colon and the caecum). Additional unpublished data was obtained from post-mortem examinations carried out by MSE and MC over the last decades. Because in many sources, the caecum was not indicated separately but as part of the LI measurement, the colon was not recorded separately. When the body mass of the individuals was not available from the source but only SVL or CL was given, body mass was estimated from SVL or CL using published allometric equations [15–17]. For *Podocnemis* spp., no corresponding equation was published; therefore, we derived our own BM-CL regression equation based on data from various sources [18–24].

Weighted species means (correcting for sample size) were calculated of each morphometrical parameter and the corresponding body mass. For example, if more data was available for small intestine than for caecum length of a species, then the body mass used for associations with small intestine length was different from the one used in the same species for associations with caecum length. Data on the natural diet of reptiles was collated from the literature; species were classified as faunivores, omnivores or herbivores. To our knowledge, no continuous measure of reptile diets is available. The referenced dataset is available as an online supplement. Mammal data used for comparison was taken from Duque-Correa et al. [3].

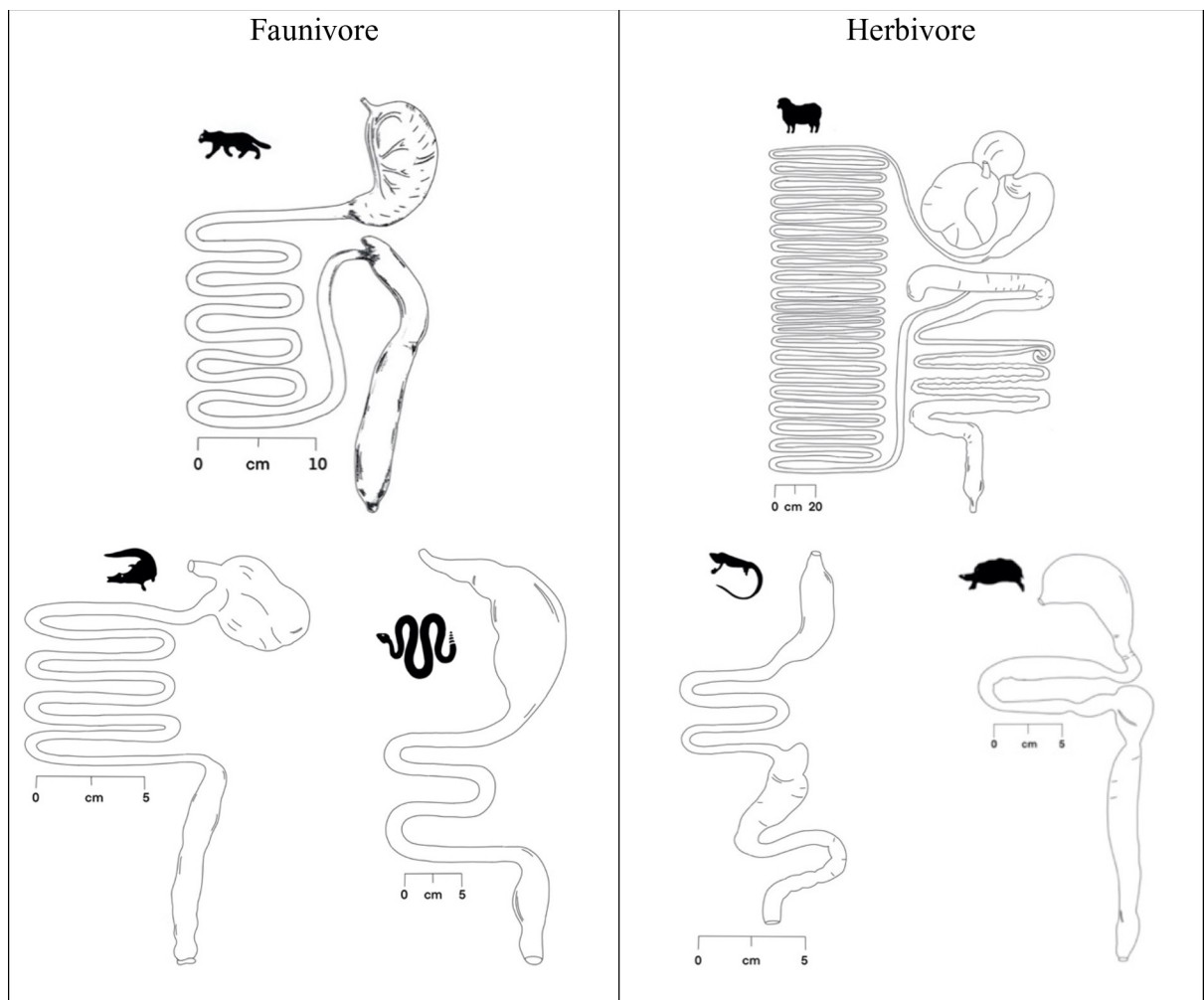

**Fig 1. Examples of digestive tracts of faunivorous and herbivorous mammals as compared to reptiles.** Modified from Stevens and Hume [9]. Note the seemingly less distinct differentiation in intestinal length across trophic groups in reptiles.

The phylogenetic tree for reptiles was built from the supertrees (based on molecular data and time calibrated by the fossil record) for Squamates [downloaded from http://vertlife.org/phylosubsets/, following 25] and for Testudines [26]. Trees were pruned in R using scripts from the library 'ape' [27] and 'tidyverse' [28] in order to obtain a final tree inclusive only of the species for which GIT data and body masses were available. For mammals, the phylogenetic tree from Duque-Correa et al. [3] was used; the reptile and mammal trees were merged manually following the amniote topology from Fig 1 in Gemmell et al. [29]. The two crocodilian species were added manually as a sister group of the testudines. Divergence times between all major manually merged clades were set using TimeTree.org, which provides an estimated time of divergence. This was also used to separate *Crocodylus* from *Alligator* (80.0 Ma). Thus, the split between mammals and sauropsids (root of the tree) was set at 318 Ma. The separation of lepidosaurids (Squamates) from the clade of Testudines and Crocodylia was set at 280 Ms while the internal split between Testudines and Crocodylia was set at 254 Ma. The emergence of Squamates was set to 180.57 Ma.

Statistical analyses were done on (i) all available data (i.e., at different sample size for the different intestine sections–generally larger samples for the total intestine than for individual

sections), and on a subset that comprised (ii) those species for which both small and large intestine length was available. Additionally, analyses were performed on the taxonomic levels of squamates, lizards, snakes, and turtles. First, the allometric relationships with body mass were determined, and it was assessed which intestine section showed the best fit with body mass. Scaling exponents were termed 'more' or 'less than geometric' if they were above or below the expected isometry of 0.33 [30]. Then, the effect of diet was evaluated.

Allometric regressions were performed as linear regressions on log-transformed data, because we are not aware of another method to which we can apply phylogenetic generalized least squares. All analyses were performed using generalized least squares (GLS) phylogenetic generalized least squares (PGLS), recording the 95% confidence interval for parameter estimates, using the R packages 'caper' [31] and 'nlme' [32]. The phylogenetic signal lambda ($\lambda$) was estimated by maximum likelihood. The significance level was set to 0.05. Different models applied to a certain dataset (separately for GLS and PGLS) were compared using the small sample corrected Akaike's information criterion ($AIC_c$) [33], considering models that differed by more than 2 ($\Delta AIC_c > 2$) as providing a different fit to the data.

## Results

Intestinal length information was available for the total intestinal tract (157 species), the small intestine (147 species), the large intestine (141 species), and the caecum (52 species). Generally, the small intestine represented the longest intestinal section, followed by the large intestine and the caecum (Fig 2). In analyses using phylogenetical generalized least squares (PGLS), the phylogenetic signal $\lambda$ varied across its whole range from 0 to 1, depending on the specific dataset; generally, larger datasets had a higher $\lambda$ (Table 1, S1-S8 Tables in S1 File).

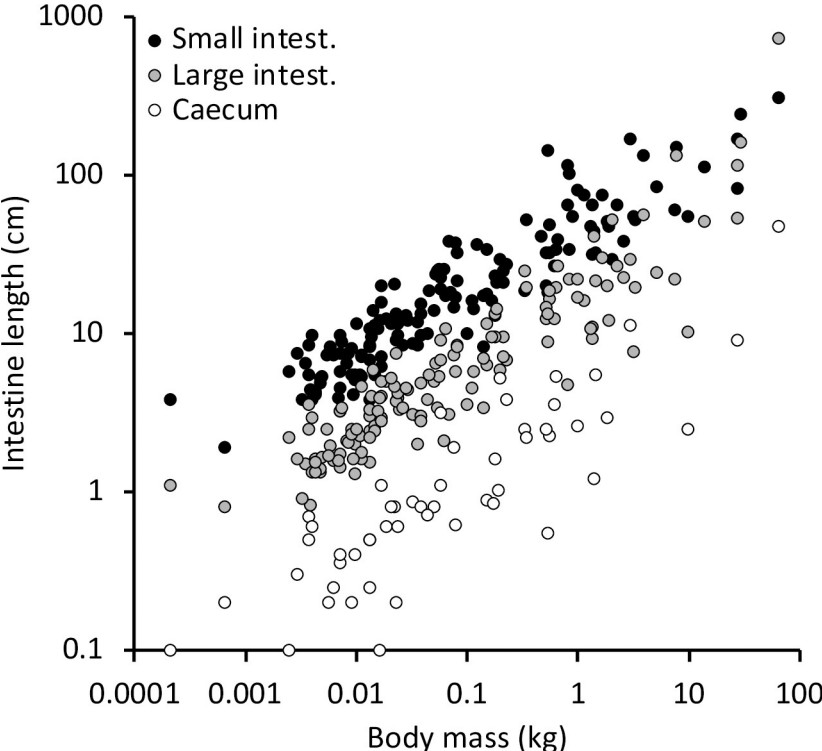

**Fig 2. Magnitude comparison of the length of the small intestine, the large intestine and the caecum in reptiles.**
Note that the slope of the scaling of the three sections in log-log-space appears similar.

**Table 1. Summary statistics for allometric scaling of insteinal length with body mass in reptiles, as log(y) = a + b log(body mass), or y = (10$^a$) BM$^b$ (significant parameters in bold).**

| Dependent | Model | n | | GLS AICc | ΔAICc | parameter (95%CI) | PGLS lambda (95%CI) | AICc | ΔAICc | parameter (95%CI) |
|---|---|---|---|---|---|---|---|---|---|---|
| *All data* | | | | | | | | | | |
| Total intest. | BM | 157 | a | - | - | **1.82 (1.79 to 1.86)** | **0.77 (0.56 to 0.90)** | - | - | **1.77 (1.61 to 1.92)** |
| | | | b | | | **0.40 (0.38 to 0.42)** | | | | **0.38 (0.35 to 0.4)** |
| Small intest. | BM | 147 | a | - | - | **1.66 (1.62 to 1.70)** | **0.50 (0.22 to 0.74)** | - | - | **1.62 (1.49 to 1.75)** |
| | | | b | | | **0.38 (0.36 to 0.41)** | | | | **0.37 (0.33 to 0.40)** |
| Large intest. | BM | 141 | a | - | - | **1.24 (1.19 to 1.29)** | **0.56 (0.31 to 0.75)** | - | - | **1.14 (0.98 to 1.31)** |
| | | | b | | | **0.43 (0.40 to 0.46)** | | | | **0.40 (0.37 to 0.44)** |
| Caecum | BM | 52 | a | - | - | **0.48 (0.37 to 0.59)** | 0.00 (NA to 0.27) | - | - | **0.48 (0.37 to 0.59)** |
| | | | b | | | **0.42 (0.35 to 0.49)** | | | | **0.42 (0.35 to 0.49)** |
| *Consistent data (species for which both small and large intestinal length are available)* | | | | | | | | | | |
| Small intest. | BM | 141 | a | -82.3 | 0.0 | **1.64 (1.60 to 1.68)** | **0.56 (0.31 to 0.75)** | -125.5 | 0.0 | **1.61 (1.46 to 1.75)** |
| | | | b | | | **0.38 (0.35 to 0.40)** | | | | **0.36 (0.33 to 0.39)** |
| Large intest. | BM | 141 | a | -26.5 | 55.8 | **1.24 (1.19 to 1.29)** | **0.57 (0.33 to 0.76)** | -61.7 | 63.8 | **1.14 (0.98 to 1.31)** |
| | | | b | | | **0.43 (0.40 to 0.46)** | | | | **0.40 (0.37 to 0.44)** |

NA no model output.

## Allometry

In most datasets analyzed, intestinal lengths scaled more-than-geometrically (positive allometry) at an exponent whose 95% confidence interval was above 0.33. Regardless of the phylogenetic signal, the simple scaling relationships were generally similar in generalized least squares (GLS) and PGLS (Table 1). When using only species for which all respective data were available, the small intestine-body mass relationship showed a better fit that the large intestine-body mass relationships (ΔAIC$_c$ GLS = 56, PGLS = 63), suggesting that the large intestine is more subjected to additional influence factors (Table 1). Body mass was part of all subsequent models.

## Taxonomic comparisons

Based on body mass, no differences in the scaling between the reptile groups (lizards, snakes, turtles) was evident (S1 Fig and S1 Table in S1 File), even though numerically, snakes had the lower and turtles the higher scaling exponents.

Compared to mammals, reptiles generally had shorter intestines at comparable body mass (Fig 3, S2 Table in S1 File). The models including only body mass but not the taxonomic group (mammal / reptile) always had the lowest support (ΔAIC$_c$ >150). Using the difference in the GLS models, reptile intestines are only 30–40% of the length of mammalian intestines at similar body mass. In GLS, both the taxon group (mammal / reptile) and the body mass-taxon group interaction were generally significant, except for the large intestine, where the interaction was not. Consequently, the model including the interaction had the best support (ΔAIC$_c$ to other models: total intestine 9.7, small intestine 10.1, caecum 17.0), except for the large intestine where the model that only included the taxon group was best supported (ΔAIC$_c$ to other models 7.0). Hence, while the scaling for the large intestine was similar between mammals and reptiles (exponent 95% confidence intervals 0.39–0.44 vs. 0.40–0.46, respectively), the scaling of the total and small intestine was steeper in mammals than in reptiles (at exponents for the total intestine of 0.46–0.49 vs. 0.38–0.42, small intestine of 0.44–0.47 vs. 0.36–

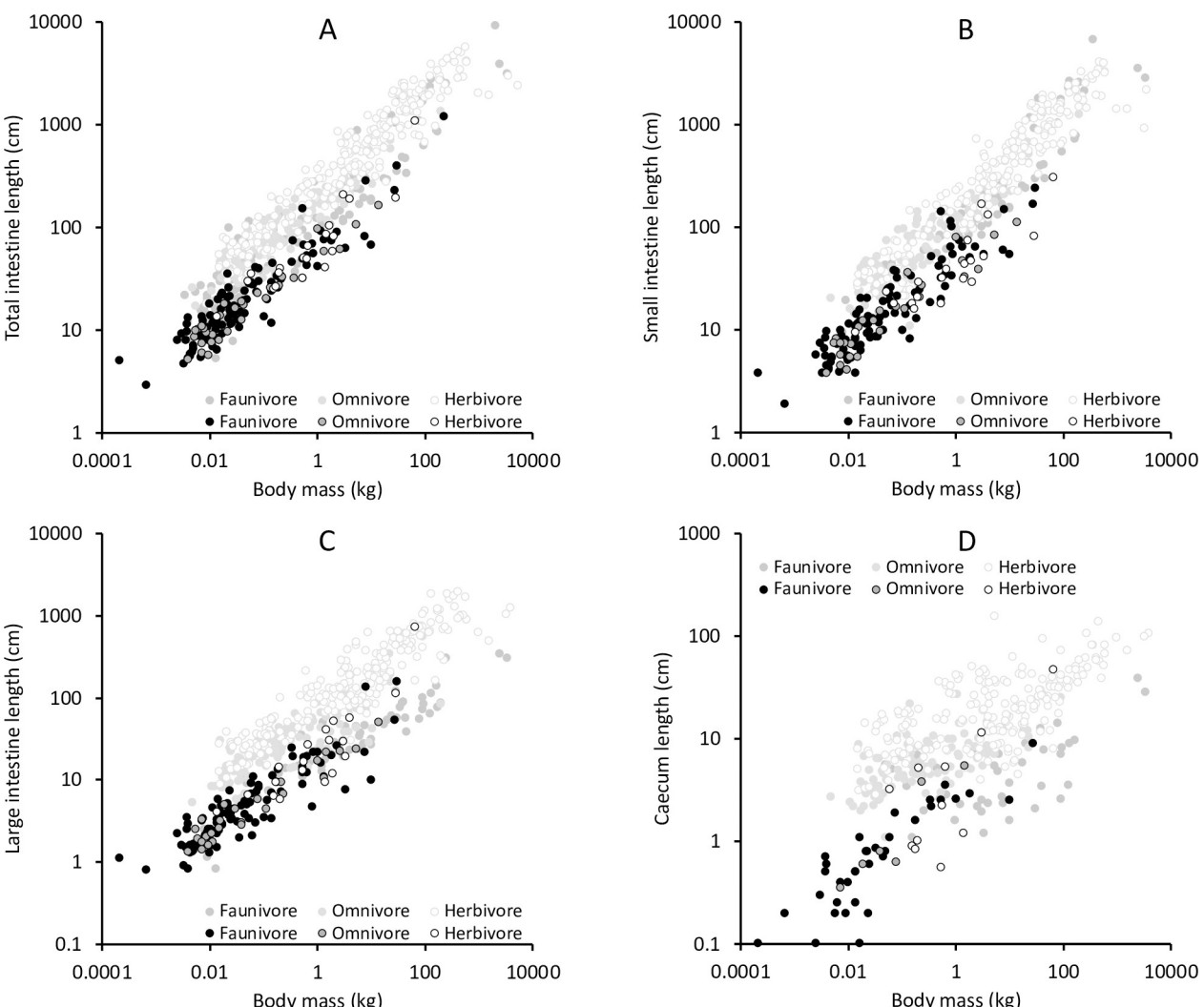

**Fig 3.** Relationship of body mass, intestinal length and trophic level in reptiles for (A) total intestine (n = 157 species), (B) small intestine (n = 147), (C) large intestine (Caecum and colon) (n = 141), (D) caecum (n = 52), as compared to mammals (light grey symbols). For statistics, see S3-S5 Tables in S1 File. Mammal data from [3].

0.41, respectively). For the caecum, scaling in mammals was lower than in reptiles (0.18–0.24 vs. 0.35–0.49, respectively).

In PGLS, there was a strong phylogenetic signal when analyzing mammals and reptiles together ($\lambda \geqq 0.90$). In PGLS, taxon group (mammal / reptile) was never significant, and the body mass x taxon group interaction was only significant for the caecum. Models that included taxon group (with or without interaction) were never better supported than the models with body mass only ($\Delta AIC_c$ always $\leqq 2$; S2 Table in S1 File).

## Trophic level

Adding the categorical information on trophic level (fauni-, omni-, herbivore) to the models for all reptiles did not result in an improved data fit in GLS, and in PGLS only in the case of the large intestine (with $\Delta AIC_c$ to the model without trophic group = 2.4; S3 Table in S1 File); visually, scatter plots did not indicate a systematic effect of trophic level (Fig 3). In GLS,

trophic level was significant for the large intestine, with herbivores having a 1.3 times longer large intestine than faunivores. This difference was also supported in PGLS (S3 Table in S1 File). When making the same assessment for only those reptile species for which both small and large intestine length was known, the result was the same. Again, the small intestine-body mass relationship had a better fit than the large intestine-body mass relationship, and the addition of trophic level did not improve model fit in GLS, and in PGLS only barely for the large intestine ($\Delta AIC_c$ to the model without trophic group = 2.4 (S3 Table in S1 File)).

When considering reptile taxa individually, adding trophic level did not improve model fits in Squamates or Testudines (only for the Squamates large intestine in PGLS, S4 Table in S1 File). However, in the lizards, it improved model fit for the total intestine (GLS $\Delta AIC_c$ = 2.9, PGLS $\Delta AIC_c$ = 8.3), and in PGLS also for the small and the large intestine ($\Delta AIC_c$ 2.8 and 3.8, respectively). In both models, herbivorous lizards had significantly longer measures than faunivorous ones (S4 Table in S1 File). Because snakes only included faunivores, they were not assessed separately for trophic level.

In the large-scale comparisons with mammals, GLS models that included both taxonomic group (mammal / reptile) and trophic level were the best supported for total intestine, large intestine and caecum ($\Delta AIC_c$ > 84), and among the best supported for the small intestine (S5 Table in S1 File). Reptiles had significantly shorter, and herbivores longer intestines in all these models. In PGLS, taxonomic group was never significant and did not improve model fit; trophic level was included in the best models for total intestine, large intestine and caecum, and herbivores always had significantly longer measures. By contrast, the best models for the small intestine did not include trophic level in PGLS (S5 Table in S1 File).

## Body mass vs. body length scaling

Given that the original concept of intestinal length being linked to diet in reptiles originated from Lönnberg [4], who did not use body mass but snout-vent-length (SVL) as the basis for the comparison, we re-analyzed this individual dataset (n = 40 species, total intestine length) with and without the trophic level information provided by that author. In both GLS ($\Delta AIC_c$ = 9.7) and PGLS ($\Delta AIC_c$ = 3.2), trophic level addition improved model fit, and while herbivores only had a significantly longer total intestine in GLS but not PGLS (95% CI of parameter estimate 0.24 to 0.58 and -0.02 to 0.46, respectively) than faunivores, omnivores had significantly longer measures than faunivores in both models (0.05 to 0.33 and 0.04 to 0.29, respectively). In these models, total intestine length scaled to SVL at an exponent that included linearity (GLS: 0.64 to 1.11, PGLS: 0.87 to 1.35).

Based on these observations, comparisons were repeated for the dataset that was based originally on SVL. In all reptiles, adding trophic level to the SVL-based GLS and PGLS models increased model fit for the total, small and large intestine, and for the caecum in PGLS ($\Delta AIC_c$ to the model without trophic level GLS > 24, PGLS > 3; S6 Table in S1 File). Models always indicated longer measurements in herbivores compared to faunivores. Models that did not include trophic level but whether a species was a snake or not had even better model fits in GLS for the total and small intestine and the caecum ($\Delta AIC_c$ to trophic level models GLS > 6) but not for the large intestine (S6 Table in S1 File), indicating that snakes are not only peculiar in being exclusively faunivores, but also either with respect to intestine length or to SVL. A dataset comprising only lizards and turtles led to identical results (except that the factor 'snake' was not assessed; S7 Table in S1 File). In the lizard only dataset, trophic level always not improved model fit ($\Delta AIC_c$ GLS > 7, PGLS > 2.6; S7 Table in S1 File). For turtles, trophic level improved model fit only in GLS for the large intestine ($\Delta AIC_c$ = 11.9; S7 Table in S1 File), with herbivores having longer measures, but not in PGLS.

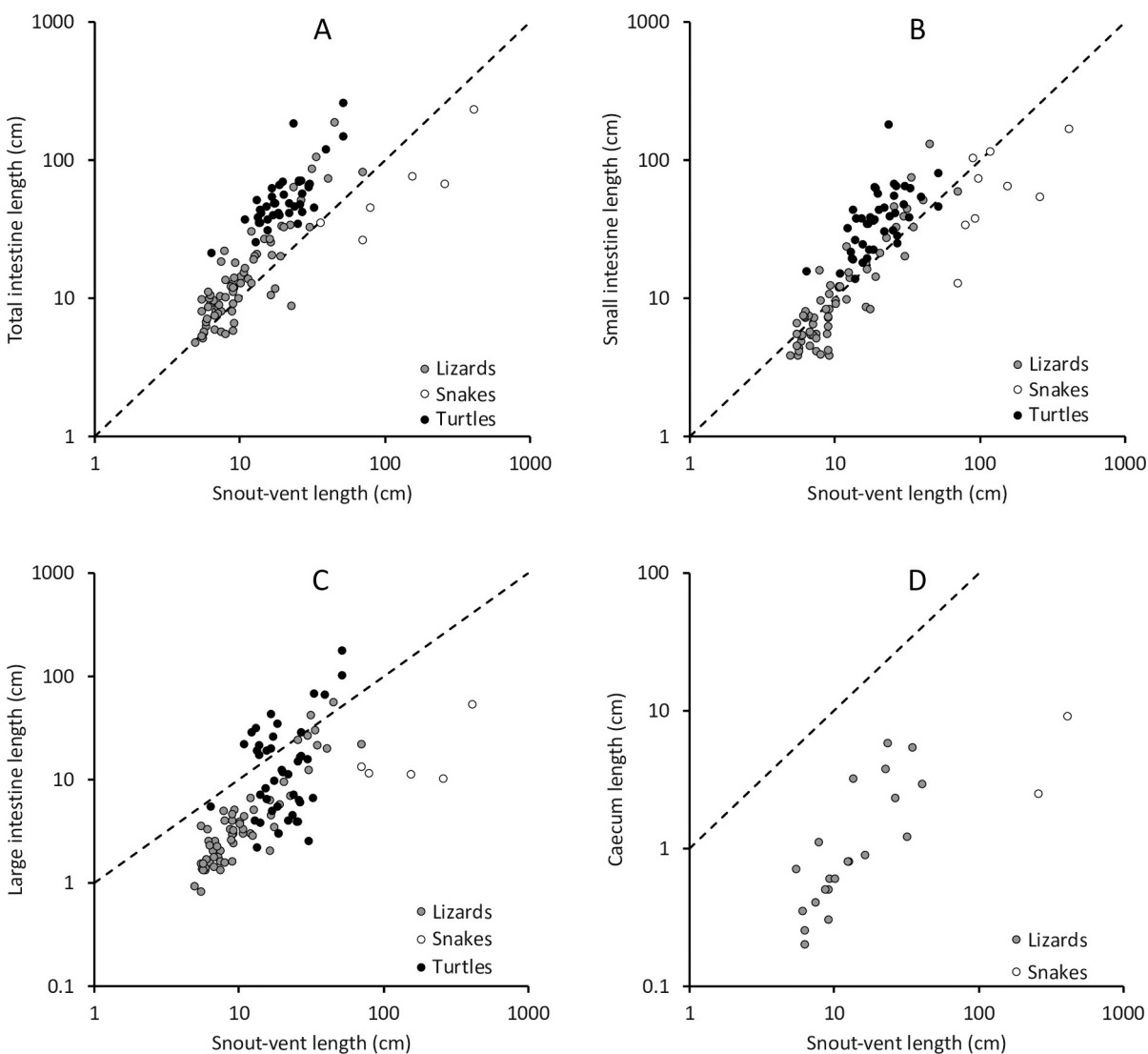

**Fig 4.** Relationship of snout-vent-length (SVL) and intestinal length in reptiles for (A) the total intestine, (B) the small intestine, (C) the large intestine, (D) caecum, for taxonomic groups. Note that snakes are all faunivorous and have, at a similar intestine length, a similar body mass as other reptiles, but a longer body length. For statistics, see S8 Table in S1 File. The dotted line represents y = x to facilitate gauging the slope of the length-length relationships.

When comparing model fit in the dataset for the total intestine length in which both body mass and SVL could be used as a basis for scaling, the body mass-based model had the better fit ($\Delta AIC_c$ GLS 63, PGLS 32), and this difference was maintained when other factors were added. This is due to the difference in the scaling in snakes, which have a similar intestine length as other reptiles for their body mass (S1 Fig in S1 File), but–predictably–a shorter intestine length than other reptiles for their body length (Fig 4). Adding being a snake or trophic level information to GLS models improved model fit similarly, both for body mass and SVL models (S8 Table in S1 File). In PGLS, being a snake did not add substantially to the model fit with body mass and trophic level ($\Delta AIC_c \lessgtr 2$); for PGLS SVL models, trophic level and being a snake both contributed substantially to model fit. In the same dataset, SVL as dependent variable scaled with body mass in a geometric fashion (with an exponent whose 95% confidence interval included 0.33); the scaling model fit was dramatically improved when being a snake

was included (ΔAIC$_c$ GLS 109, PGLS 38), so that at the same body mass, snakes had a 4–4.5 times higher SVL than other reptiles. Trophic level did not improve this scaling (S8 Table in S1 File).

## Discussion

The present study provides a comprehensive data collection on reptilian intestinal length, which indicates that a distinct association with trophic niche is not as evident in reptiles as in mammals. The results show that it is important to consider different sections of the intestinal tract separately when possible, most likely because they fulfil different functions. Even though our data collection represents possibly the largest one available to date, presented also in the hope that it will be useful for other researchers, the limitations of both the dataset itself and the focus on a single kind of measurement–length measures–must be stated.

### Constraints

Compared with mammals [3], distinctively fewer data on intestine length exist in reptiles that can be linked, at least, to snout-vent- (SVL) or carapax length (CL). Large collections on reptiles kept as pets either did not resolve their sample to the species level [34], or did not record a proxy (body mass, CL for turtles) that would make the data comparable [35]. Meiri [16] already argued that reptile studies should more often report the body mass of the investigated specimens, a claim which we support. Generating more data on reptile gastrointestinal morphology should be a feasible task, considering the occurrence of reptiles in zoological institutions or as pets.

Gastrointestinal dimensions and tissue mass can fluctuate distinctively in reptiles, especially in faunivorous species with an intermittent feeding regime [36, 37], or in animals undergoing seasonal dormancy [38]–factors that could not be accounted for in the present data collection. As in similar large-scale comparisons in mammals [3, 39], variation due to origin (free-ranging or captive), inter-individual differences due to actual diets consumed, tissue preservation or body condition could also not be accounted for. For example, already the original study by Lönnberg [4] does not indicate the origin, the actual diet, feeding or dormancy state of the specimens, nor the means of their preservation.

As already mentioned by Lönnberg [4], SVL may not represent an ideal proxy for body size because of the difference in body shape (slender vs. stout) between species. Transforming SVL or CL into body mass could dampen or exaggerate potential differences between trophic levels, if trophic level had an effect on the body mass-SVL relationships. This was not the case in the present dataset (S8 Table in S1 File). However, Meiri [35] reported the nearly significant effect (at $P = 0.058$) that herbivorous lizards were heavier for their SVL, or had a shorter SVL for their body mass–in other words, they had a stouter body shape. When estimating body mass from SVL using a general equation that does not account for trophic level, this would mean that herbivorous animals would be assigned a slightly underestimated body mass, which should make any putatively longer intestinal tract even more evident in comparative approaches. The fact that this was not the case suggests that the result of little large-scale trophic differentiation in intestinal length of the present study is robust in spite of the SVL-body mass transformation.

### Allometry

As described for different mammalian datasets [reviewed in 3], all intestinal lengths investigated in the present study scaled at a higher exponent than expected based on simple geometry (i.e., positive allometry at an exponent > 0.33). The only consistent exception to this were the

snakes, whose intestinal scaling exponent always included 0.33 in the confidence interval (S1 Table in S1 File).

The current explanation for the positive allometry is that whereas gut surface scales geometrically with body mass, diffusion distance (i.e., gut diameter) should remain short, i.e. scale less-than-geometrically (with negative allometry), and hence gut length must necessarily scale more-than-geometrically to compensate [40]. This hypothesis would have to be tested in two ways: (i) with a large dataset on intestinal diameter; (ii) with a large dataset on intestinal surfaces, which would test the underlying assumption that this surface scales geometrically with body mass. Neither dataset exists or can be collated easily, to our knowledge, and thus resolution of these question hinges on the future measurement of anatomical data.

Based on the concept of geometric gut surface scaling, a possible reason for the lower scaling in snakes could lie in the very intermittent feeding pattern in the snakes [41, 42] with a particular dependence on gastric dilatation, but little dependence on a long intestine with a high throughput capacity.

## Comparison with mammals

The present study impressively corroborated previous observations based on smaller sample sizes [8–10] that reptiles have shorter digestive tracts than mammals. This is explained by the lower metabolism of the poikilothermic reptiles as compared to homeothermic mammals [43, 44]. At least for herbivorous reptiles, both the wet gut contents and the calculated dry gut contents are of a similar magnitude as in mammals [11, 45], suggesting a similar capacity of the main digesta-holding gut compartments. Together with longer intestines in mammals, this would necessarily imply intestines of larger diameter in reptiles. Again, to our knowledge, a comparative dataset on intestinal diameter does not exist. On the other hand, in a comparison of two reptile and two mammal species, Karasov et al. [8] reported a lesser nominal (i.e., without villi) surface of the small intestine in the reptiles, suggesting that mammal intestines are not only longer to compensate for putatively smaller diameter, but actually have a larger surface and hence larger volume capacity. Without further measurements, this discrepancy remains unresolved.

On the basis of comparable body mass, reptiles have lower food intakes than mammals [8, 45]. Assuming a similar gut volume, this necessarily translates into a slower passage of digesta through the digestive tract [8, 45]. Mammals, with their higher energetic requirements, need to eat more, and consequently have a faster digesta passage. To avoid negative effects of this faster passage on digestive efficiency, mammals employ several strategies in parallel: they achieve a distinct particle size reduction during ingestion via chewing [46], as smaller particles can be digested at a much faster rate than larger ones [reviewed in 47]. They possibly have shorter diffusion distances in the intestines via smaller intestine diameters (as deducted from the similar gut capacity and the longer intestines demonstrated in the present study). And they possibly have more absorptive surface due to microanatomical structures, in particular in the small intestine; their intestine also achieves higher absorption rates [48], and their intestinal functions are evidently less dependent on ambient temperatures. Because of these (and probably other) adaptations, mammals achieve a similar and even higher digestive efficiency than reptiles in spite of their higher intakes [8, 45, 49, 50].

Whereas these considerations refer to the difference in the scaling factor (the 'intercept' in log-linear regressions), we also found a lesser scaling exponent (the 'slope' in log-linear regressions) for the total and the small intestine in reptiles as compared to mammals [3] (e.g., small intestine: GLS exponent 0.36–0.41 in reptiles vs. 0.44–0.47 in mammals). In PGLS, this difference was not significant (as indicated by the non-significant taxon-body mass interactions in

S2 Table in S1 File) (small intestine: PGLS exponent 0.33–0.40 in reptiles as compared to 0.38–0.43 in mammals). This indicates that the seeming difference in slope is caused by the distribution of specific taxa across the dataset but not by a systematic difference between reptiles and mammals. By contrast, the difference in the scaling exponent for the caecum was also significant in PGLS (GLS exponent in reptiles 0.35–0.49 vs. 0.18–0.24 in mammals; PGLS in reptiles 0.35–0.49 vs. 0.25–0.32 in mammals). As previously, we suggest that this is due to particularly long caeca in small mammals, possibly associated with the widespread strategy of 'colonic separation mechanism'-linked coprophagy in this group [3], and not due to a particularly steep scaling in reptiles (in which such a mechanism and the associated coprophagy have not been reported).

Individual studies comparing organ masses between selected mammal and reptile species of similar body mass indicate larger organ masses in mammals [e.g. 51, 52]. Larger literature compilations of data from several studies show that, for a similar body mass, reptiles have less gastrointestinal tissue [11; this study]. Additionally, while liver mass is probably similar in mammals and reptiles, heart and kidney mass might be slightly lower in reptiles [11]. This raises the interesting question what it is that reptiles have 'more' than mammals to achieve the 'similar body mass'. To our knowledge, this question has not been answered so far, and we can only speculate that reptiles may have distinctively more body tissue in the form of their tails.

## Trophic level

The general argument is that a less digestible diet, typically one of a higher proportion of plant material, requires a longer digestive tract [1, 53]. More specifically, in mammals it was the large intestine, not the small intestine, that showed this adaptation to a plant diet [3]. In his original study on lizards, Lönnberg [4] already described that it is particularly the large intestine that was longer in herbivorous species. By contrast, Herrel et al. [54] found that it was particularly the small intestine that was longer in four omnivorous vs. eleven faunivorous lizards, and hypothesized that faunivory (as opposed to any proportion of plant matter in the diet) should be particularly distinguishable by intestine length in lizards.

For reptiles, a large number of studies described particular adaptations to herbivory. These studies typically focus on the caeco-colonic dilatation and valve-like structures in lizards [4, 6, 54–56], but intestinal length has also been used [7, 57–62]. However, our data collection does not indicate clear differences in intestinal tract length between the trophic groups. The only indication was that the large intestine was significantly longer in herbivores by a factor of 1.3 (S3 Table in S1 File), compared to a GLS-derived factor of 1.9 in mammals [3]. The difference between GLS and PGLS in the reptiles suggests that whereas the pattern is not distinct when just comparing the trophic groups (Fig 2), the difference between the trophic groups is systematic within various taxa. In this dataset, these taxa belonged to the lizards. When lizards were analyzed alone, the longer large intestines (but not small intestines) in herbivorous species were detectable in both GLS and PGLS (S5 Table in S1 File).

In the dataset based on SVL measurements only, trophic level had an effect on intestinal lengths in the complete dataset, even when the taxonomic group of snakes was accounted for in the models, mainly due to a large intestine that was 1.4 times longer in herbivores than in faunivores (S6 and S7 Tables in S1 File). In the SVL dataset, the number of turtle species that could be included was distinctively higher than in the BM dataset (63 vs. 18 species). This was due to one investigation [35] that did not record BM or carapax length in turtles, but only SVL, making a transformation to BM impossible (because no transformation equations for SVL to BM exist for turtles). The fact that larger datasets lead to stronger trophic signals was already observed in mammals [3]. Therefore, data for more reptile species might well lead to a

stronger signal on trophic specialization of intestinal length. Based on the available data, however, we conclude that a trophic signal in reptile intestinal length is weak.

We can only speculate about reasons for a putatively less distinct effect of trophic level on intestine length in reptiles than in mammals. Although to our knowledge, a comparative dataset is missing that assesses the relative food intake, digesta passage and digestibility in herbivorous vs. faunivorous reptiles, we assume that, as in mammals [63], herbivorous reptiles should generally have a higher food intake than faunivorous ones. However, with reptile energy expenditures [44] and food intake levels [45] of approximately only 10% of that of mammals, the basic reptilian intestinal tract might not be as challenged as that of mammals with respect to optimization of digestion *and agility*. A lack of requirements for agile locomotion might contribute to the difference in mammals. Possibly, a lower proportion of faunivorous reptiles are active pursuit hunters, with ambush predation considered the ancestral state [64]. Insectivorous forms employ a sit-and-wait or a search foraging strategy [65, 66], and snakes that include relatively large vertebrate prey in their spectrum are typically sit-and-wait predators [67]. In mammals, the clear difference in intestinal tract length [3] as well as in abdominal cavity volume [14] between herbivores and faunivores may well derive from an arms race in agility [13, 68–70], where in particular the faunivores reduced intestinal tissue and body shape to maximize pursuit efficiency. Possibly, reptiles additionally move less within a comparable time frame than mammals [12] in spite of similar home range sizes [71]. Comparative evaluations of reptile and mammal movement and hunting patterns, as well as comparative datasets on digestive performance of reptilian herbivores and faunivores, are required to further test these hypotheses. The less differentiated reptile gastrointestinal anatomy might also correspond to the finding that the gastrointestinal microbiome shows a less stringent phylogenetic signal in reptiles than in mammals [72].

## Body shape

The finding that intestine length is similar across reptile groups when compared to body mass but not to snout-vent-length is due to the special position of the snakes (Fig 4). Clearly, the elongation in body length in snakes is not accompanied by a corresponding elongation of the intestine, even though other organs, such as the stomach, the kidneys, lungs or spleen clearly appear elongated when compared to other reptiles [73]. This contrasts with the mustelids–a group of mammalian carnivores that also evolved an elongated body shape [74]. At comparable body mass, mustelids have longer intestines than other carnivores [75], mirroring a comparison of body mass and body length [76]. Thus, the hypothesis results that different mechanisms of body elongation occur in snakes and mustelids–resulting in different effects on the intestinal tract. More studies, including embryological ones, are required to address this difference.

## Conclusions

The results of the present study indicate that while a link between trophic niche and intestinal length can be detected in some datasets, and in particular for the large intestine, this link appears weaker in reptiles compared to mammals, which might be linked to a lesser need for differentiation in reptiles because of their lower metabolism, lower food intake, and less agile movement patterns. Clearly, more data are required to fully address these questions. In particular, future studies could fruitfully combine a variety of measurements, not only intestinal lengths, but organ tissue and contents mass, diameters and surfaces, as well as both body mass and snout-vent-length or carapax length, to facilitate more comprehensive comparisons that have to rely less on inductive speculation.

## Supporting information

**S1 File. Supplemental information.** S1-S8 Tables & S1 Fig.
(DOCX)

**S2 File. Phylogenetic tree.** Reptiles & mammals.
(TXT)

**S3 File. Reptile intestine data.** Full dataset.
(XLSX)

**S4 File. Reptile intestine data.** Lönnberg dataset.
(XLSX)

**S5 File. *Podocnemis* equation.** Data for regression line.
(XLSX)

**S6 File. Data sources.** Reference list.
(DOCX)

## Acknowledgments

We thank Barbara Schneider and Jacqueline Wick for tireless support in literature acquisition. The contributions to this project by the Disease Investigations and Animal Care staff at San Diego Zoo Global, Charles Paddock Zoo and Pacific Wildlife Care, as well as numerous student assistants, are greatly appreciated. We thank Emilia Clauss for drawing Fig 1, Dan Rabosky for spotting an important mistake in a previous analysis, and reviewers and editors for comments on the manuscript.

## Author Contributions

**Conceptualization:** Marcus Clauss.

**Data curation:** Monika I. Hoppe, Carlo Meloro, Mark S. Edwards, Marcus Clauss, María J. Duque-Correa.

**Formal analysis:** Marcus Clauss, María J. Duque-Correa.

**Funding acquisition:** Marcus Clauss.

**Investigation:** Monika I. Hoppe, Marcus Clauss, María J. Duque-Correa.

**Methodology:** Carlo Meloro, Daryl Codron, Marcus Clauss.

**Project administration:** Monika I. Hoppe, Marcus Clauss, María J. Duque-Correa.

**Resources:** Mark S. Edwards, Marcus Clauss.

**Supervision:** Marcus Clauss, María J. Duque-Correa.

**Visualization:** Monika I. Hoppe, Marcus Clauss.

**Writing – original draft:** Monika I. Hoppe, Marcus Clauss, María J. Duque-Correa.

**Writing – review & editing:** Carlo Meloro, Mark S. Edwards, Daryl Codron.

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
