## [Decision Letter · Decision Letter 0]

18 May 2021

PONE-D-21-11487

Less need for differentiation? Intestinal morphology of reptiles as compared to mammals

PLOS ONE

Dear Dr. Clauss,

Thank you for submitting your manuscript to PLOS ONE. After careful consideration, we feel that it has merit but does not fully meet PLOS ONE’s publication criteria as it currently stands. Therefore, we invite you to submit a revised version of the manuscript that addresses the points raised during the review process.

I am happy if general morphological-physiological trends in animals are being analyzed, However, there are few strong conclusions derived from your data. Please look at the comments below. Hope you can improve the manuscript. 

We look forward to receiving your revised manuscript.

Kind regards,

Ulrich Joger

Academic Editor

PLOS ONE

Additional Editor Comments:

The reviewer I found for your manuscript found nothing “wrong” in this paper with methods or statistics, and the writing appeared clear. Yet he proposed to reject it. I am not so hard and like to give you a Chance to improve the paper. Please enlarge the data (if you have more) or Change your conclusions by carefully differentiating between hard evidence and speculation. It is not a shame if a hypothesis is not supported by a data set, but a discussion must be self-critical.

Journal Requirements:

 [This study is part of Swiss National Science Foundation (http://www.snf.ch) project CRSII5_189970 / 1 (to MC). The funders had no role in study design, data collection and analysis, decision to publish, or preparation of the manuscript.]. 

Reviewers' comments:

Reviewer's Responses to Questions

**Comments to the Author**

1. Is the manuscript technically sound, and do the data support the conclusions?

Reviewer #1: Partly

2. Has the statistical analysis been performed appropriately and rigorously? 

Reviewer #1: Yes

3. Have the authors made all data underlying the findings in their manuscript fully available?

Reviewer #1: Yes

4. Is the manuscript presented in an intelligible fashion and written in standard English?

Reviewer #1: Yes

5. Review Comments to the Author

Reviewer #1: This manuscript analyzes and compares in reptiles and terrestrial mammals variation in intestine length in relation to body size, diet, and phylogeny. The statistical analysis of the data seem fine, and the writing is clear.

My main critique is that the paper advances knowledge about digestion or ecomorphology very little because it focuses on one thing, really one dimension, intestine length. As a reader, I approach it with about as much interest as I would a thorough analysis of some other intestinal analogue such as intestine width. Width could be analyzed in the same way as length is here, but who would be interested in that? Put the two together and one would certainly have a lot more, such as surface area that can be related to substrate absorption or niche space for adherent microbes, or gut volume that can be related to holding capacity for digesta or microbes. Add time (another dimension) and one can make inferences about turnover kinetics and the matching of rates such as digesta breakdown or metabolic demand. But working with gut length (or width) alone limits saying much definitive about function. The authors recognize this (i) by pointing out that most authors who consider herbivorous reptile gut ecomorphology focus on the caeco-colonic dilatation and valve-like structures (lines 369-371), and (ii) with statements that additional data on gut diameter, which are necessary to make stronger inferences related to gut length, are not available (line 299,314-320, 329). Consequently, there are few conclusions I would call reliable, some seem poorly supported by data, and many points of discussion seem thin and speculative.

For example, this seems to be the case for every point discussed in the section “Comparison with mammals” (lines 306-358). Because the issue of differences in gut diameter is unresolved, the authors try to infer it using an inconclusive argument (lines 310-320) followed finally by invoking the idea that mammals do have smaller intestine diameters (lines 329-330), which they do not have strong evidence for.

As another example, they speculate about differences in scaling exponent of intestine length (beginning line 38), which is perhaps not even meaningful without knowing intestine diameter and hence actual surface area. According to Karasov and Hume (1997) there is no difference in the scaling exponent of intestine nominal surface area between reptiles (5 species) and mammals (20 species). That’s not a lot of evidence either, so if the science needs more data then the best solution is to provide more data.

At lines 353-358 – I do not see why the authors state that liver size is probably similar in mammals and reptiles. The data from Else and Hulbert (1985) suggest otherwise, as do the regression lines shown in Fig. 1c of Franz et al. (2009). The authors finish this section with the sentence “we can only speculate that reptiles may have distinctively more body tissue in the form of their tails”.

Testing putative trophic differences in intestine length was the “raison d’etre” for the paper (lines 26-28; 51-55). In the Discussion of this (lines 360-365) they point out that in mammals it applies mainly to the large intestine, which they also cite as Lonnberg’s conclusion for reptiles >100 years ago. It is what the authors found again here, mainly in lizards. The advance in knowledge that they offer is that the trophic signal might be weaker in reptiles than in mammals, though “larger datasets lead to stronger trophic signals” (lines 390-391). So, they end this section of the Discussion with speculation “…about reasons for a putatively less distinct effect of trophic level on intestine length in reptiles than in mammals” (395-396).

I reiterate that I found nothing “wrong” in this paper with methods or statistics, and the writing is clear. I mainly fault it for its narrow focus that to my thinking leads to conclusions poorly supported by data, and little advance in more reliable understanding, as opposed to speculation.

6. PLOS authors have the option to publish the peer review history of their article (what does this mean?). If published, this will include your full peer review and any attached files.

Reviewer #1: No

---

## [Author Response · Author response to Decision Letter 0]

21 May 2021

please see the detailed reply letter

---

## [Editor Report · Decision Letter 1]

31 May 2021

Less need for differentiation? Intestinal morphology of reptiles as compared to mammals

PONE-D-21-11487R1

Dear Dr. Clauss,

We’re pleased to inform you that your manuscript has been judged scientifically suitable for publication and will be formally accepted for publication once it meets all outstanding technical requirements.

Kind regards,

Ulrich Joger

Academic Editor

PLOS ONE

Additional Editor Comments (optional):

Your answers to the critique by the reviewer convinced me that there is not really much you can do to improve your manuscript - only add some words in the discussion explaining what the limits of your data are and what should be done in the future. As you have done this to my satisfaction, I accept your manuscript now. Just put genus names like Crocodylus and Alligator in italics.
---

## [Editor Report · Acceptance letter]

25 Jun 2021

PONE-D-21-11487R1 

Less need for differentiation? Intestinal length of reptiles as compared to mammals 

Dear Dr. Clauss:

I'm pleased to inform you that your manuscript has been deemed suitable for publication in PLOS ONE. Congratulations! Your manuscript is now with our production department. 

Kind regards, 

on behalf of

Dr. Ulrich Joger 

Academic Editor

PLOS ONE